# Infodemiologists Beware: Recent Changes to the Google Health Trends API Result in Incomparable Data as of 1 January 2022

**DOI:** 10.3390/ijerph192215396

**Published:** 2022-11-21

**Authors:** Pieter Hermanus Myburgh

**Affiliations:** Metaverse Research Unit, Institute for Intelligent Systems, University of Johannesburg, Johannesburg 2092, South Africa; hermanm@uj.ac.za

**Keywords:** online search engine activity, Google Trends, freebase ID, Google knowledge graph identifiers, infodemiology, pharmacosurveillance, Google Trends for flu, social science research

## Abstract

In an ever-increasingly online world, many Internet users seek information from online search engines such as Google. Accessing such search activity allows infodemiologists a glimpse into the collective online mind. Tools such as Google Trends and Google Health Trends (GHT) can be used to gauge search activity in key geographical regions and for specific periods of time. Recently, Google implemented changes to the GHT platform. Evidence is provided here for an initial exploration of how this change impacted the data obtained from GHT. Comparing 177 weekly probabilities for short search sessions of 421 Freebase IDs in thirty geographies extracted from GHT both before and after the implemented change, a low correlation (median of all Spearman ρ = 0.262 [IQR 0.04; 0.53]) between these data was observed for the year 2022. In general, the extracted values are higher after the implemented changes, compared to the values extracted before the change. Future research using the GHT API should not attribute increases in GHT data from 1 January 2022 onward as being reflective of increased search activity for a specific keyword, but rather attribute it to the implemented change to the GHT sampling strategy.

## 1. Introduction

Global access to the worldwide web has increased remarkably, with just over 63% of the global population accessing the Internet in 2022 [1]. Although access to the search giant Google is limited in some regions (such as China and North Korea), it dominates search activity in most other territories (Table 1). Knowing what the world is searching for online gives researchers the opportunity to identify and respond to these trends in a timely manner. As such, gaining access to search activity has long been regarded a holy grail for researchers, with different tools used in the assessment of such patterns.

Specific to Google’s search engine, the unrestricted Google Trends platform (https://trends.google.com/, accessed on 22 September 2022) is open for all to explore how specific demographics searched for certain keywords. Google Trends retrieves a relative search volume (RSV), a metric ranging from 0 to 100 and based on the proportional popularity of the keyword in a specific geographic region for the selected period. Although this platform gives the user an indication of the dates or times that a specific phrase was searched the most frequently, it lacks the ability for users to compare results from different periods [2]. For example, searching the same keyword for different time periods on the same geographical boundary yields different results (Figure 1). Since this is a scaled metric, based on the number of searches in the geographical limitation selected for the searched keyword, comparisons between regions are not possible with Google Trends data [3]. The extraction of Google Trends data can be automated, to some extent, using unofficial application programming interfaces (APIs), such as pytrends (v4.8.0) for Python [4].

For those interested in comparing search activity in different regions or time periods, Google offers limited access to the Google Health Trends (GHT) API. Access can be requested from https://bit.ly/3xpYFJo. GHT was used to explore the search behavior of African Internet users related to the COVID-19 pandemic, as a prediction tool for dengue fever in Brazil, and to gauge interest in pre-exposure prophylaxis in the United States of America [5,6,7], with mixed reports on the effectiveness of this tool in infodemiology and epidemiology.

Recently, Google announced via email that the GHT API “will be improved by providing higher precision responses by using a more comprehensive sample of search requests” (Appendix A). The said changes were implemented on 18 July 2022, with all data from 1 January 2022 being altered to include this new comprehensive search request sample. Google also indicated that any changes in search interest dating 1 January 2022 might be attributable to this change.

Such changes impact ongoing research, especially when future research efforts seek to compare periods before and after the implementation of such a change. The GHT documentation has also not been updated yet to indicate that such a change was made, risking the potential that erroneous conclusions can be made in the future. Here, I present an investigation into whether this change implemented by Google indeed had an impact on the GHT data retrieved and provide the first evidence that future research using the GHT platform should refrain from comparing data obtained from 1 January 2022 onwards to dates before 2022.

## 2. Materials and Methods

### 2.1. Data Extraction from the Google Trends API

The use of Freebase IDs, or in the absence of a Freebase ID, the corresponding Google Knowledge Graph Identifiers (GKGIs), allows searching for specific terms regardless of the searcher’s input language, since Google aggregates search values based on these identifiers. For example, searches conducted for ‘watre’ (sic), ‘水’, ‘l’eau’, ‘जलम्‌’, ‘metsi’ or ‘amanzi’ would be categorized as a search for ‘/m/0838f’ corresponding to the English word ‘water’. Freebase IDs or GKGIs were identified using the Google Knowledge Graph Search API and used as search terms on the GHT API, according to the recommendation by Google. Using Freebase IDs, therefore, allows for comparable search data across linguistically different searches.

The presented study was based on two different datasets extracted from the GHT API. First, the probabilities of short search sessions of 421 Freebase IDs (Appendix A) were searched in 30 countries (Table 1) before the recent update to the Google Trends sampling strategy. These extractions were carried out between 9 and 12 June 2022 for a different research project, the author having no prior knowledge of the pending change in the GHT random sampling strategy. A second extraction was performed after the changes were made to the GHT API on 22 July 2022. Weekly probabilities for short search sessions were extracted for the period from 6 January 2019 to 22 May 2022, resulting in 177 weeks’ worth of data extracted for each of the searched terms in all countries. The extractions were carried out using a Python script as per Google’s guidelines [8]. The only modification was that the process was automated by including for loops to conduct the extractions for different countries.

### 2.2. Statistical Analyses

Statistical analyses were performed in R (R Core Team, v4.2.0, 2022), using RStudio Integrated Development for R. The raw data extracted were plotted as two separate time series, applying locally estimated scatterplot smoothing (LOESS) to visually identify potential trends. Spearman correlation was used to determine the correlations between data obtained from the two data extractions and summarized. Thereafter, a new time series was constructed by calculating the difference between the values retrieved via the Google Trends API before and after the updates occurred on 18 July 2022. These time series of differences were also plotted. Anomalies (datapoints that are outside the normal fluctuation range of a time series) in the different time series were detected using the AnomalyDetection package for R [8] and the anomaly time series were plotted using the internal plotting functions of R, as well as ggplot2 [9].

## 3. Results

In total, 12,630 time series were extracted both before and after the implemented change to the Google Trends API, plotted with the application of LOESS and visually inspected for potential trends. These figures are publicly available here: https://doi.org/10.25415/ujhb.20424642.v2. Visual inspection was indicative of a high degree of similarity between the extracted data points from 2019–2021, with divergences in general trends occurring more frequently in the data from 2022 onward (Figure 2 as an example).

For the extracted timeframes, a high correlation was observed between the data extracted before and after the update for the years 2019–2021, with respective median correlation values [interquartile range] of 0.955 [0.93; 0.98], 0.961 [0.93; 0.98] and 0.956 [0.93; 0.98] for these years (Figure 3). However, for the first months of 2022, the median correlation for the 421 included search terms was much lower, 0.262 [0.04; 0.53].

The difference was then calculated and plotted for the extracted data. These figures are made available publicly at: https://doi.org/10.25415/ujhb.20424693.v1.

Since 177 data points (corresponding to weekly search activity) were extracted from the Google Trends API for each time series, a total of 2,235,510 data points were included in this study, of which ~7.42% (165,953) were identified as anomalies using the AnomalyDetection package for R. Plots of data points identified as anomalies in the difference plots are made publicly available here: https://doi.org/10.25415/ujhb.20430924.v1. Anomalies in a constructed difference time series occur due to Google’s daily updates of the uniformly distributed random sample of searches from which the data are extracted. As such, some variance is expected, as was the case in anomalies detected for 2019–2021 (Table 2). However, most (79.40%) of the anomalies detected in the collected data originated in 2022. The median values of these anomalies between the two extractions were similar for 2019–2021, while the median for 2022 was double that of previous years.

Within the 30 countries included in this investigation, all returned an increased number of anomalies in the 2022 data, ranging between 46.09% (China) and 96.18% (India) of anomalies in these time series (Table 3).

## 4. Discussion

The Google Trends API gives researchers the ability to access search trends from most countries around the world. Little is known regarding the sampling strategy that Google implements to construct the GHT database, apart from the statement in the GHT API Getting Started Guide: *“Numbers are calculated on a uniformly distributed random sample of Google web searches done since 2004, updated once a day, thus there may be some variance between similar requests”* [10].

As such, fluctuations in data retrieved on different extraction days are expected. Although such variance can affect data for a specific search term on a specific day, general trends in time series have a high correlation between data extracted on different days. From the two data sets extracted before and after the changes were made to the Google sampling strategy, a high degree of correlation was observed for the data extracted for 2019–2021 (Figure 3). This is in line with the notification received on the changes made to the sampling strategy. In its email, Google indicated that the changes to the sampling strategy will only affect data from 1 January 2022 onward (Appendix A).

These changes in the sampling strategy resulted in a greater range of correlation values between older and newer data sets for the year 2022 to date (Figure 3A,E), as well as a lower median correlation value. The low similarity between the data extracted before and after the change in sampling strategy is indicative of the implemented change to the data used to retrieve the Google Trends data. By detecting anomalies in the difference between these two time-series, we were able to show that changes implemented to the GHT sampling strategy mostly increased the returned values (Table 1), with the median value of these unexpected differences in 2022 being double the value of previous years. In the 30 countries investigated, the majority of unexpected data points from the differenced time series occurred in 2022 (Table 2). Through a visual inspection of the plotted time series, most search terms showed an increasing trend during the first months of 2022.

Since this newly implemented change to the sampling strategy results in predominantly higher search volume being returned, data extracted prior to 18 July 2022 can no longer be compared to data extracted after this date for the year 2022. However, the high level of correlation for previous years is indicative that, in most cases, comparative studies focused on dates prior to 1 January 2022 could still be accurate considering the minor variance introduced by Google’s daily updates to the sample data set. As mentioned elsewhere [11], caution should be exercised in the interpretation of single extractions of GHT API data, which may be falsely interpreted as changes in search trends. Therefore, it is advised that the extractions of the GHT API data be repeated on different dates and analyzed accordingly.

The presented study was not without limitations. Owing to the short timeframe between the announcement that the GHT sampling strategy will be changing and the date of implementation of these changes, only data from a singular extraction prior to the implemented change could be analyzed. It is also uncertain as to which increases were due to the changes made to GHT, or which were attributable to the chance of the GHT sample dataset on the day of extraction. Although this limits the quantification of the changes made to the sampling algorithm, the results are indicative that the changes impacted the data obtained from the service, that there is mostly an increase in search probability for most search terms after 1 January 2022, and that the interpretation of comparative studies with data extracted after the implemented changes should be handled with caution.

## 5. Conclusions

Evidenced here is the first report that the recent changes to the sampling strategy implemented by Google impacted the comparability of the GHT API data, particularly on comparisons of search trends from before and after January 1, 2022. Although the improved sampling strategy may result in a more accurate representation of search trends, caution should be exercised on any increased search trends observed following the 1 January 2022 date and extracted after the 18 July 2022. Furthermore, it would be impossible to determine whether such changes indeed gave a more representative view of the use of the Google Search Engine by individuals. Although such changes may impact current research activities involving the GHT API, the improved sensitivity that may arise from this change and the benefits of having an improved GHT API may, in the future, result in better predictions—which could be especially useful when using the Google Trends API for public health monitoring.

## Figures and Tables

**Figure 1 ijerph-19-15396-f001:**
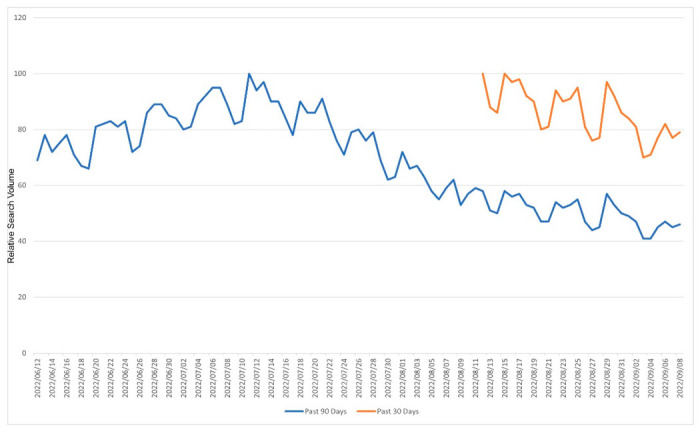
Google Trends results for ‘Coronavirus disease 2019′, retrieved on 12 September 2022 from the Google Trends platform (https://trends.google.com/) based on Worldwide web searches. The data retrieval period was set to the ‘past 90 days’ (blue line) and the ‘past 30 days’ (orange line).

**Figure 2 ijerph-19-15396-f002:**
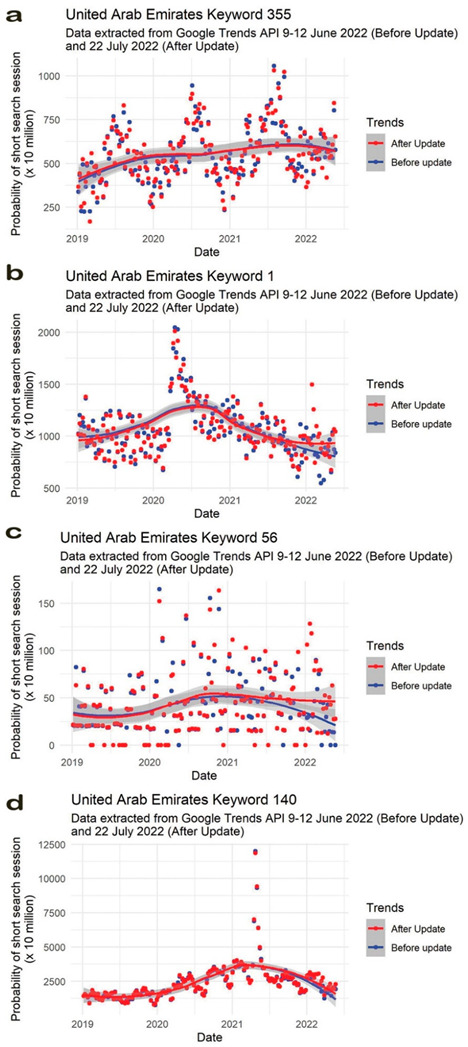
Example plots for four (**a**–**d**) arbitrarily selected search terms in the United Arab Emirates. LOESS was applied to visually identify potential trends. In most plots, a divergence occurs between the smoothed time series after 2022. The 12,630 figures are publicly available here: https://doi.org/10.25415/ujhb.20424642.v22.

**Figure 3 ijerph-19-15396-f003:**
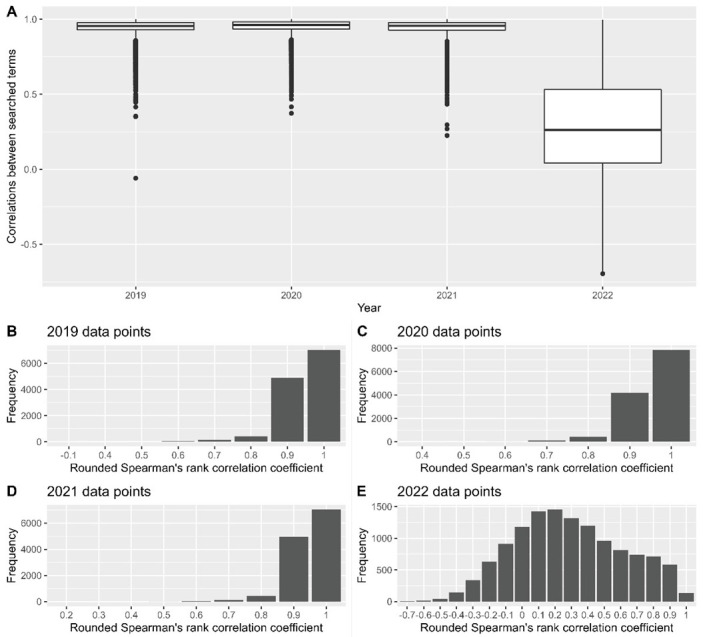
Summary of Spearman’s rank correlation coefficients between 421 terms in 30 countries before and after the implementation of changes to the Google Trends API. (**A**–**D**) are indicative of highly correlated values for data extracted for 2019–2021, with low levels of correlation in 2022 (**E**).

**Table 1 ijerph-19-15396-t001:** List of countries included in the comparison of pre- and post-implemented changes to the Google Health Trends API.

ISO 3166-1 Alpha-2 Country Code	Country Name	Google Search Market Share ^1^
AU	Australia	93.37%
BE	Belgium	92.33%
BR	Brazil	96.29%
CA	Canada	91.18%
CN	China	3.56%
EG	Egypt	97.48%
ET	Ethiopia	91.74%
FR	France	90.76%
DE	Germany	89.96%
HK	Hong Kong SAR China	91.64%
IN	India	98.59%
IR	Iran	99.49%
IT	Italy	94.50%
JP	Japan	75.91%
MX	Mexico	93.99%
NL	The Netherlands	93.26%
NZ	New Zealand	93.21%
NG	Nigeria	98.30%
PE	Peru	96.31%
ZA	South Africa	91.60%
KR	South Korea	69.58%
ES	Spain	94.47%
CH	Switzerland	90.94%
TH	Thailand	98.47%
UG	Uganda	96.50%
AE	United Arab Emirates	96.13%
GB	United Kingdom	91.74%
US	United States	86.99%
UY	Uruguay	95.65%
ZW	Zimbabwe	93.35%

^1^ Market share as of June 2022. Data extracted from https://gs.statcounter.com/search-engine-market-share/all/ (accessed on 4 June 2022) under Creative Commons Attribution-Share Alike 3.0 Unported License.

**Table 2 ijerph-19-15396-t002:** Proportional distribution and basic descriptive summaries of anomalies detected in time series constructed from two retrievals of Google Trends API data; before and after the implemented change to the Google Sampling strategy.

	Anomalies 2019 *n* (%)	Anomalies 2020 *n* (%)	Anomalies 2021 *n* (%)	Anomalies 2022 ^1^ *n* (%)
Smaller probability after the update (% within year)	4263 (34.23)	3499 (33.17)	3780 (33.78)	31,570 (23.96)
Larger search probability after the update (% within year)	8190 (65.77)	7049 (66.83)	7410 (66.22)	100,192 (76.04)
Total anomalies per year (% between years)	12,453 (7.50)	10,548 (6.36)	11,190 (6.74)	131,762 (79.40)
Median [IQR]	20.10 [−18.37; 110.30]	20.20 [−16.09; 126.20]	19.10 [−17.23; 87.90]	40.00 [6.16; 114.50]

^1^ until 22 May 2022.

**Table 3 ijerph-19-15396-t003:** Anomalies observed in the time series comparing the difference in Google Trends API data collected before and after the implemented changes to the data set.

Country	2019–2022 Total Anomalies *n*	2019 Anomalies *n* (%)	2020 Anomalies *n* (%)	2021 Anomalies *n* (%)	2022 Anomalies *n* (%) ^1^
Australia	4807	121 (2.52)	149 (3.10)	206 (4.29)	4331 (90.10)
Belgium	5186	299 (5.77)	293 (5.65)	404 (7.79)	4190 (80.79)
Brazil	5662	86 (1.52)	80 (1.41)	86 (1.52)	5410 (95.55)
Canada	4835	118 (2.44)	130 (2.69)	163 (3.37)	4424 (91.50)
China	6617	1187 (17.94)	1521 (22.99)	859 (12.98)	3050 (46.09)
Egypt	5570	398 (7.15)	301 (5.40)	388 (6.97)	4483 (80.48)
Ethiopia	6817	1096 (16.08)	886 (13.0)	1059 (15.53)	3776 (55.39)
France	5069	127 (2.51)	177 (3.49)	209 (4.12)	4556 (89.88)
Germany	5214	114 (2.19)	102 (1.96)	180 (3.45)	4818 (92.41)
Hong Kong SAR China	5803	593 (10.22)	436 (7.51)	431 (7.43)	4343 (74.84)
India	6118	14 (0.23)	90 (1.47)	130 (2.12)	5884 (96.18)
Iran	4658	554 (11.89)	260 (5.58)	201 (4.32)	3643 (78.21)
Italy	5316	155 (2.92)	133 (2.50)	203 (3.82)	4825 (90.76)
Japan	4848	168 (3.47)	174 (3.59)	175 (3.61)	4331 (89.34)
Mexico	5925	87 (1.47)	90 (1.52)	119 (2.01)	5629 (95.00)
The Netherlands	5175	202 (3.9)	204 (3.94)	280 (5.41)	4489 (86.74)
New Zealand	5422	444 (8.19)	381 (7.03)	548 (10.11)	4049 (74.68)
Nigeria	5580	611 (10.95)	353 (6.33)	389 (6.97)	4227 (75.75)
Peru	5342	360 (6.74)	250 (4.68)	279 (5.22)	4453 (83.36)
South Africa	4643	370 (7.97)	313 (6.74)	261 (5.62)	3699 (79.67)
South Korea	5076	620 (12.21)	335 (6.60)	343 (6.76)	3778 (74.43)
Spain	5136	116 (2.26)	158 (3.08)	165 (3.21)	4697 (91.45)
Switzerland	4864	344 (7.07)	280 (5.76)	380 (7.81)	3860 (79.36)
Thailand	5166	275 (5.32)	233 (4.51)	223 (4.32)	4435 (85.85)
Uganda	6828	1251 (18.32)	924 (13.53)	773 (11.32)	3880 (56.82)
United Arab Emirates	5296	571 (10.78)	391 (7.38)	389 (7.35)	3945 (74.49)
United Kingdom	5317	93 (1.75)	106 (1.99)	196 (3.69)	4922 (92.57)
United States of America	6457	81 (1.25)	95 (1.47)	130 (2.01)	6151 (95.26)
Uruguay	6304	833 (13.21)	660 (10.47)	945 (14.99)	3866 (61.33)
Zimbabwe	6902	1165 (16.88)	1043 (15.11)	1076 (15.59)	3618 (52.42)

^1^ Up to 22 May 2022.

## Data Availability

This communication originated from within another ongoing research project. The list of keywords searched on the Google Health Trends API is included as Appendix A. All figures used as part of this investigation are made available on the University of Johannesburg’s data repository under the following DOIs: https://doi.org/10.25415/ujhb.20424693.v1, https://doi.org/10.25415/ujhb.20430924.v1, and https://doi.org/10.25415/ujhb.20424642.v2.

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
