# Peer review of "Infodemiologists Beware: Recent Changes to the Google Health Trends API Result in Incomparable Data as of 1 January 2022"

_ijerph, 2022, doi:10.3390/ijerph192215396_

Round 1

Reviewer 1 Report

While the finding supports the claim, It is rather challenging to determine whether all the increases or some of the increases are solely responsible for the sampling changes implemented by GHT. Perhaps some discussion on this would be a nice contribution. 

Author Response

Dear Reviewer

Thank you for your review of this submission.  I value your time reading through this manuscript and your suggestions for improving it.  

I agree with the comment made that there should be more discussion on the limitation that the increases observed could potentially be due to chance.  The following paragraph has been added to the discussion:

The presented study was, however, not without limitations. Due to the short timeframe between the announcement that the GHT sampling strategy will be changing and the date of implementation of these changes, only data from a singular extraction prior to the implemented change could be analyzed. It is also uncertain as to which increases were due to the changes made to GHT, or which were attributable to the chance of the GHT sample dataset on the day of the extraction.  Although this limits the quantification of the changes made to the sampling algorithm, the results are indicative that the changes impacted the data obtained from the service, that there is mostly an increase in search probability for most search terms after 1 January 2022, and that the interpretation of comparative studies with data extracted after the implemented changes should be handled with caution.   

I value any additional comments to improve the manuscript.
Sincerely, 

Reviewer 2 Report

The paper discusses an important topic of data credibility and how changes in Google Health Trends API lead to an inflated GHT search activity and cautions against it.

Abstract

“Comparing 177 weekly probabilities for short search sessions of 421 Freebase IDs in thirty geographies extracted from GHT both before and after the implemented change, a low correlation (median of all Spearman ρ = 0.262 [IQR 160.04; 0.53]) between these data.” The sentence seems to be incomplete. Please review and update.

Introduction

·         Line 30 not “on-line” change to online

·         Line 56 it should be pre-exposure not “preexposure”

General Comments  - If the author can clarify a bit more about what an anomaly is, that would improve the quality of the manuscript.  

Additional comments:

1. What is the main question addressed by the research? The research aims to investigate the impact of the new changes in Google Health Trend implemented since early 2022 on yielding search results. 
2. Do you consider the topic original or relevant in the field? Does it address a specific gap in the field? The topic is original. However, the topic's importance is more cautionary than critical, creative research. 
3. What does it add to the subject area compared with other published material? This adds a piece of critical information and provides caution to future researchers regarding how to interpret Google Health Trends (GHT), which, if not considered, could appear inflated. 
4. What specific improvements should the authors consider regarding the methodology? What further controls should be considered? The author uses a short and limited dataset to draw a conclusion about the research question. Hence the author should consider expanding the dataset and address this as a limitation. Further, currently, a simple correlation test is performed, a stronger statistical test should be performed to evaluate if old and new methods vary significantly. 
5. Are the conclusions consistent with the evidence and arguments presented, and do they address the main question posed? Yes, a stronger statistical analysis could strengthen the paper. 
6. Are the references appropriate? Yes.
7. Please include any additional comments on the tables and figures. Figure 1 -  No Y-axis

Author Response

Dear Reviewer
Thank you for your time and dedication towards improving this manuscript.  I am highly appreciative.   

As per your review, I've made the following changes:

Abstract
Comparing 177 weekly probabilities for short search sessions of 421 Freebase IDs in thirty geographies extracted from GHT both before and after the implemented change, a low correlation (median of all Spearman ρ = 0.262 [IQR 0.04; 0.53]) between these data was observed for the year 2022.

Line 30
...for online gives researchers the opportunity to identify and respond to these trends in a... 

Line 56
...fever in Brazil, and to gauge interest in pre-exposure prophylaxis in the United States of...

Figure 1

y-axis added (Relative Search Volume)

General comments

"Anomalies" were described as follow:
Anomalies (datapoints that are outside the normal fluctuation range of a timeseries) in the difference time series were detected using the AnomalyDetection... 

"Hence the author should consider expanding the dataset and address this as a limitation."

I want to thank the author for this suggestion.  I agree that expanding the dataset to include other terms may have indeed resulted in a more accurate study.  The challenge, however, is that the implemented change happened unexpectedly and without much warning.  It is, therefore, impossible to increase the dataset of searched terms, as the change was already implemented, and we cannot obtain data on search terms before this change.  I have improved the discussion with the addition of the following paragraph:

The presented study was not without limitations. Due to the short timeframe between the announcement that the GHT sampling strategy will be changing and the date of implementation of these changes, only data from a singular extraction prior to the implemented change could be analyzed. It is also uncertain as to which increases were due to the changes made to GHT, or which were attributable to the chance of the GHT sample dataset on the day of extraction.  Although this limits the quantification of the changes made to the sampling algorithm, the results are indicative that the changes impacted the data obtained from the service, that there is mostly an increase in search probability for most search terms after 1 January 2022, and that the interpretation of comparative studies with data extracted after the implemented changes should be handled with caution.   

"Further, currently, a simple correlation test is performed, a stronger statistical test should be performed to evaluate if old and new methods vary significantly. ...Yes, a stronger statistical analysis could strengthen the paper. "
Again, a very valid suggestion.  Although I agree with the reviewer that in some cases, correlation may be seen as a "weaker" statistical test, I am of the opinion that in this instance Spearman's rank correlation is an applicable and reliable test for the data presented.  The compared time series (two at a time) are before and after the implemented change to GHT.  Correlation was tested for each of the 421 searched terms in 30 countries.  As such, the focus is on how well-aligned the two data sets are.  Since each data point correlated between the two datasets is for the same day, there should be a high correlation between datasets (as is seen for 2019 - 2021 data).  This is not the case for data points from 2022, where a large difference is observed between datasets extracted from GHT.  
That said, I would be more than willing to run additional statistical analyses based on the recommendation of statistical tests by the reviewer.  The ultimate goal of this manuscript is to warn GHT users that the changes did impact the comparability of data from 1 January 2022 onward, which can be seen in the summary presented here by means of correlation analyses.

Again, I want to express my sincere gratitude towards the reviewer for their work to improve this manuscript, and I look forward to further bettering the result.

Kind regards

Reviewer 3 Report

The paper is interesting, but it is necessary to make a reference to other studies on the subject.

It is necessary to have the state of the art, and the extensive bibliographic list. Indeed, during the pandemic period, working from home required more frequent use of the Internet, which led to various studies like the one presented in the paper.

Author Response

Dear Reviewer
Thank you for your support and improvements to this manuscript.  I greatly appreciate your time.  

Per your recommendation, the manuscript was edited by a language practitioner at the University of Johannesburg.  These changes were implemented in the resubmission.  The document is presented in US English, and will also be edited as part of the IJERPH publishing process.

Regarding the increase in literature requested, there are reference to three articles that was published recently on the topic in the literature.  The presented work is a short communication, to alert the scientific community of the implemented changes.  Since this format is aimed at quickly describing a change or reporting preliminary results, I believe the three mentioned examples of the use of GHT to be sufficient.  That said, I would be happy to add another specific example which could be recommended by the reviewer.

Thank you once more for your suggestions and your improvements to the manuscript.  It is greatly appreciated.  I shall endeavour to make any additional changes that you see fit to ensure that the manuscript is worth publishing. 

Kind regards